# Motion Analysis of Triangular Fibrocartilage Complex by Using Ultrasonography Images: Preliminary Analysis

**DOI:** 10.3390/s22010345

**Published:** 2022-01-04

**Authors:** Issei Shinohara, Atsuyuki Inui, Yutaka Mifune, Hanako Nishimoto, Kohei Yamaura, Shintaro Mukohara, Tomoya Yoshikawa, Tatsuo Kato, Takahiro Furukawa, Yuichi Hoshino, Takehiko Matsushita, Ryosuke Kuroda

**Affiliations:** Department of Orthopaedic Surgery, Graduate School of Medicine, Kobe University, 5-2, Kusunoki-cho 7, Chuo-ku, Kobe-shi 650–0017, Japan; 203m878m@stu.kobe-u.ac.jp (I.S.); ainui@med.kobe-u.ac.jp (A.I.); hanako-nishi@live.jp (H.N.); koheidesuyo@yahoo.co.jp (K.Y.); no.8-shintaro@hotmail.co.jp (S.M.); tomo_yoshi_0926@yahoo.co.jp (T.Y.); tkato@med.kobe-u.ac.jp (T.K.); taka1023@med.kobe-u.ac.jp (T.F.); u1ho4no@med.kobe-u.ac.jp (Y.H.); matsushi@med.kobe-u.ac.jp (T.M.); kurodar@med.kobe-u.ac.jp (R.K.)

**Keywords:** TFCC, ultrasonography, articular disc, PIV, dynamic analysis

## Abstract

The triangular fibrocartilage complex (TFCC) is a significant stabilizer of the distal radioulnar joint. Diagnosing TFCC injury is currently difficult, but ultrasonography (US) has emerged as a low-cost, minimally invasive diagnostic tool. We aimed to quantitatively analyze TFCC by performing motion analysis by using US. Twelve healthy volunteers, comprising 24 wrists (control group), and 15 patients with TFCC Palmer type 1B injuries (injury group) participated. The US transducer was positioned between the ulnar styloid process and triquetrum and was tilted ulnarly 30° from the vertical line. The wrist was then actively moved from 10° of radial deviation to 20° of ulnar deviation in a 60-rounds-per-minute rhythm that was paced by a metronome. The articular disc displacement velocity magnitude was analyzed by using particle image velocimetry fluid measurement software. The mean area of the articular discs was larger on ulnar deviation in the control group. The mean articular disc area on radial deviation was larger in the injury group. The average articular disc velocity magnitude for the injury group was significantly higher than that for the control group. The results suggest that patients with TFCC injury lose articular disc cushioning and static stability, and subsequent abnormal motion can be analyzed by using US.

## 1. Introduction

The triangular fibrocartilage complex (TFCC) is an important stabilizer of the distal radioulnar joint (DRUJ), serving as a cushion against axial loading in the ulnar carpal joint [1]. TFCC injuries are the most common causes of ulnar wrist pain. Clinical findings include swelling, pain, limited range of forearm supination–pronation, and instability of the DRUJ with wrist pain that interferes with daily activities, such as turning door handles or shaking hands [2]. The TFCC mainly consists of four structures (Figure 1): the articular disc, meniscus homologue, radioulnar ligaments, and extensor carpi ulnaris (ECU). The ECU and the articular disc are reported to be important stabilizers of the DRUJ [1]. TFCC injury is caused by axial pressure and torsion of the wrist joint. The Palmer classification, which is widely used in clinical practice to classify TFCC injuries [1], is divided into traumatic class 1 and degenerative class 2, each of which is divided into their respective subtypes. Among them, Palmer type 1B TFCC injury occurs at the attachment of the ulnar fovea, leading to instability of the DRUJ resulting in pain [2]. The attachment to the fovea, termed as the ligamentum subcruentum (Figure 1), has been reported in recent biomechanical studies to contribute significantly to the stability of the DRUJ [3].

The TFCC is a complex soft tissue structure which cannot be evaluated by simple X-ray or computed tomography [4]. Therefore, magnetic resonance imaging (MRI) and computed tomographic arthrography (CTA) are used for the diagnosis of TFCC injuries (Figure 2); however, it is not easy to diagnose damage within the TFCC due to its small, thin, and complex structure [4]. According to a meta-analysis that analyzed the utility of different imaging modalities for TFCC injury, the sensitivities were 0.76 and 0.89 for MRI and CTA, respectively, while the specificities were 0.82 and 0.89, respectively [4]. Ultrasonography (US) has recently been used as a low-cost, minimally invasive option for the diagnosis of musculoskeletal disorders. Wu et al. presented a standard US scanning protocol for TFCC in 2019 by observing the volar and dorsal sides of the wrist joint [5]. Furthermore, US can depict components of the TFCC and elaborate existing lesions of the articular disc, meniscus homologue, and juxta-articular ligaments. However, the protocol in 2019 was described only for static image analysis [5], and dynamic analysis has not yet been performed. Previous MRI analyses have shown that the TFCC’s morphology is different between healthy volunteers and patients with Palmer type 1B TFCC injuries (radial and ulnar deviation, respectively) [6]. Therefore, we proposed that US-based observation of this morphological change would lead to the diagnosis of TFCC injury.

Although US images can dynamically assess soft tissue conditions in real time, they are subject to anisotropy and evaluator proficiency limitations [7]. Therefore, in addition to the morphological change of TFCC, we focused on the velocity of movement as a quantitative measure of instability [8]. In fact, it has been reported in cadaveric studies that dissection of the fovea attachment in TFCC causes instability. In this study, we used particle image velocimetry (PIV) as a dynamic analysis tool. PIV is a quantitative flow visualization tool that was developed for measuring fluid velocities over a wide range of lengths and timescales [9]. The PIV method was first used in fluid engineering and has since been adapted in the medical research field. Kawanishi et al. reported the use of PIV analysis in US imaging to evaluate the gliding property of the femoral fascia [10]. This technique was also used for the quantitative evaluation of the median nerve in the long-axis direction [11].

Currently, MRI and CTA are the main methods used to diagnose TFCC injuries, but these examinations are time-consuming and expensive. US imaging is a diagnostic tool that can evaluate the morphology of TFCC in real time. In this study, we hypothesized that TFCC injury can be diagnosed more efficiently by quantitatively evaluating the morphological changes shown by MRI in the past [6]. In this study, we quantitatively evaluated the morphological changes in Palmer 1B type TFCC injuries by using US images, and investigated the usefulness of US images as a diagnostic tool in clinical practice.

## 2. Materials and Methods

### 2.1. Population

Twelve healthy volunteers (nine men and three women; mean age, 36.2 years) with 24 wrists served as the control group for this study. Fifteen patients (nine men and six women; mean age, 41.3 years) with 15 TFCC injuries (Palmer type 1B) who underwent TFCC repair between April 2020 and June 2021 were included in the injury group. The diagnosis of TFCC injury was made by using MRI and intraoperative arthroscopic findings. TFCC injury cases other than Palmer type 1B were excluded. The sample size was determined by power analysis based on data from the pilot study, using G*Power 3.1. Prior sample size calculations showed that a difference in area of 2 mm^2^ was detectable in the two groups with a sample size of 30 participants (15 participants in each group), using the *t*-test (effect size = 0.95, α = 0.05, power = 0.8).

### 2.2. Experimental Data Acquisition and Analysis

On US imaging, the TFCC was visualized from its dorsal side with the forearm in pronation, using a 15M linear probe (Canon Aplio 300, TUS-A300, Canon Medical Systems, Tochigi, Japan). The gain, dynamic range, and frame rate were kept constant throughout all measurements and were not changed between participants. Following the procedure reported by Wu et al. [5], the US transducer was positioned between the ulnar styloid process and the triquetrum, parallel to the ECU tendon, with a 30-degree ulnar tilt from the vertical line (Figure 3). In this view, a hyperechoic ECU tendon could be observed as the most superficial structure. The ulnar collateral ligament could then be seen as a hyperechoic fibrillary structure beneath the ECU tendon. The lunate and triquetrum could be seen on the distal side of the view and the ulnar styloid process on the proximal side; each was covered by an anechoic layer of cartilage. The articular disc appeared as a hypoechoic structure between the ulna and lunate. The lunotriquetral ligament, which links the lunate and triquetrum, also appeared as a hypoechoic structure on the surface of two carpal bones. Between the triquetrum and the articular disc, the hammock-shaped meniscus homolog could be seen as a mixture of hyperechoic and hypoechoic structures (Figure 3).

To perform motion analysis, TFCC visualization was performed with the forearm in pronation. Under these views, the wrist was actively moved from 15 degrees of radial deviation to 25 degrees of ulnar deviation at a 60-rounds-per-minute rhythm that was paced by a metronome (Figure 3). The same examination was performed twice per wrist. All US images were evaluated by two examiners (both senior orthopedic surgeons). As described previously, the articular disc was detected as a hypoechoic area [5]. Its area of measurement was defined by four lines as follows: proximal end of the articular disc, distal end of the articular disc, ulnar collateral ligament, and joint space between the lunate and triquetrum (Figure 4a). The area was calculated by using ImageJ software (USA National Institutes of Health [NIH], Bethesda, MA, USA), which is a public-domain Java-based image-processing software developed at the United States NIH [12]. Briefly, the areas of the defined articular disc were manually plotted and measured by using the Image J measurement tool (Figure 4b). Then, a comparison of the area of the articular disc between radial and ulnar deviation was performed. Furthermore, a correlation between patient height and the mean area of their articular disc was calculated. Measurements were performed by three orthopedic surgeons, using Image J. To confirm reproducibility, we calculated the intra-rater correlation coefficient and the inter-rater correlation coefficient by using these measurement results. Using the same images, the velocity magnitude of the articular disc displacement or ECU motion was analyzed by using PIV fluid measurement software (PIV lab. Version 2.36, add-in software from MATLAB (Mathworks, Natick, MA, USA)) [9]. Briefly, two regions of interest were set at the articular disc and ECU tendon (Figure 5). We edited all US movies into 30 static images per second. The pixel displacement between two sequential images was measured, and the velocity magnitude of the structure inside the region of interest was subsequently calculated. For the articular disc, the u-component, which comprises the longitudinal movement of the articular disc, was calculated. The v-component, representing the overall axial movement of the articular disc, was also calculated. Since the u- and v-components show positive and negative values depending on the direction of movement, the evaluation was added to the absolute values. PIV measurements were performed twice, with three examiners performing the measurements. The inter-rater and intra-rater correlation coefficients were evaluated.

The relationship between the articular disc area and patient height was evaluated by using Pearson’s correlation coefficient. The area comparison of the articular disc by radial and ulnar deviation within each group and the comparison between the groups were performed by using the *t*-test. The *t*-test was also used for magnitude velocity comparison, using the PIV method. Analyses were performed by using Statcel, an Excel add-in statistical software package (Ekuseru-Toukei 2015; Social Research Information Co., Ltd., Tokyo, Japan). Statistical significance was set at *p* < 0.05. The intraclass correlation coefficient (ICC) was calculated by using R studio (version 3.6.1; R studio, Boston, MA, USA), based on the measurement results of the area and velocity magnitude. The intra-rater reliability and inter-rater reliability were also evaluated via ICC. The Ethics Committee of our institute approved this study (No. B21009), and informed consent was obtained from all patients involved.

## 3. Results

For the control group, the mean area of the articular disc was 25.37 ± 2.74 mm^2^ on radial deviation and 27.66 ± 2.89 mm^2^ on ulnar deviation. The areas during ulnar deviation were statistically larger than those during radial deviation in all cases (*p* < 0.01, *p* = 0.00009). The disc area was correlated with height in all volunteers of the control group (correlation coefficient, 0.75; *p* = 0.000135). For the injury group, the mean area of the articular disc was 29.55 ± 4.07 mm^2^ on radial deviation and 26.65 ± 3.98 mm^2^ on ulnar deviation. In contrast to the control group, the areas during radial deviation were statistically larger than those during ulnar deviation in patients with Palmer type 1B TFCC injury (*p* = 0.018) (Figure 6). For this assessment, the inter-rater reliability was 0.79, and the intra-rater reliability was 0.79. The average velocity magnitude of the articular discs, when measured by PIV, was 1.95 ± 0.48 mm/s for the control group and 3.21 ± 1.38 mm/s for the injury group. Representative images of the PIV measurement of the articular disc data for the control and injury groups are shown in Figure 7. The average velocity magnitude of the articular disc was significantly higher for the injury group (*p* = 0.011), and that of the ECU tendon was 1.94 ± 0.54 mm/s for the control group and 2.45 ± 0.49 mm/s for the injury group. The velocity magnitudes of both the articular disc and the ECU tendon were significantly higher in the injury group than in the control group (*p* = 0.042). Additionally, the average of the u-component of the articular disc was 1.55 ± 0.42 mm/s for the control group and 3.34 ± 1.07 mm/s for the injury group, showing a significantly higher u-component in the injury group (*p* = 0.00057). The average of the v-component of the articular disc was 0.75 ± 0.31 mm/s for the control group and 0.46 ± 0.09 mm/s for the injury group, showing no significant difference between the two groups (Figure 8). However, the vector directions during radial and ulnar deviation were different between the control and injury groups. The control group moved toward the direction of the US probe during ulnar deviation, whereas the injury group moved toward the direction of the US probe during radial deviation (Figure 7). This result is consistent with the measurement of the area of the articular disc. The inter-rater reliability was 0.89, and the intra-rater reliability was 0.97 for this assessment.

## 4. Discussion

In this study, we focused on the morphological changes of TFCC and investigated the usefulness of motion analysis of US images in the clinical diagnosis of TFCC injury by quantifying the area and movement velocity of the articular disc. Our results suggest that patients with Palmer 1B TFCC injury lose both cushioning and static stability of their articular disc and that subsequent abnormal motion can, in fact, be analyzed by using US.

The diagnosis of TFCC injury remains difficult in clinical settings, since no single modality has demonstrated perfect sensitivity and specificity. Currently, MRI and CTA are mainly used for diagnosis. Their sensitivities are reported to be 0.76 and 0.89, respectively, and their specificities are reported to be 0.82 and 0.89, respectively. The diagnostic accuracy differs depending on the TFCC injured site. It has been reported that the sensitivity is 0.92 and the specificity is 0.93 for central lesions, while the sensitivity is 0.71 and the specificity is 0.98 for peripheral lesions [13,14,15]. According to the reported literature, the diagnosis of peripheral injuries of the TFCC by using MRI, MR arthrography, or CTA has been less sensitive and specific than that for central injuries [16]. In addition, these diagnostic modalities are costly, and arthrography is invasive.

In recent years, quantitative evaluations of musculoskeletal disorders by using US have been reported. Chiba et al. reported an association between meniscal extrusion and the severity of knee osteoarthritis (OA) (Kellgren–Lawrence grade 3 or 4) [17]. They observed the medial meniscus of a patient diagnosed with OA on plain radiographs in the same position as that by US. The amount of deviation of the meniscus from the line connecting the medial tibiofemoral cortical surface in the US image was then measured. As a result, they reported a correlation between the amount of meniscus deviation and the progression of OA of the knee. Based on this report, we also used US to measure the area of the articular disc, which was defined by the four lines mentioned above (proximal end of the articular disc, distal end of the articular disc, ulnar collateral ligament, and joint space between the lunate and triquetrum).

Regarding the evaluation of the soft tissue around the wrist, there have been several reports using US examination for dynamic analysis of the ECU tendon. It has been reported that tendon subluxation can be observed by dynamic US examination of the ECU in patients with complaints of wrist pain [18,19]. However, the only reported diagnostic method for TFCC using US is that of Wu et al. [5]. They reported that all structures of the TFCC can be visualized by US observation from the palmar and dorsal sides. In the present study, with reference to this prior protocol, the area of the articular disc was measured by using the dorsal approach, which is useful for observing the articular disc and ECU.

From our results, the area of the articular disc was statistically larger under ulnar deviation in the healthy volunteer control group. In contrast, the area of the articular disc was statistically smaller with the wrist in ulnar deviation in the patient injury group. The articular disc plays an important role in the wrist, which bears compressive loads through its central portion. In a kinetic study using MRI, it was reported that the articular disc is elongated along its long axis by a distraction force during radial deviation and along its short axis by a compression force during ulnar deviation [6,20]. Considering our results, in the control group, the articular disc deviated toward the ulnar side, due to compression load on the disc when the wrist moved with ulnar deviation. On the other hand, the cushioning function was disrupted in the injury group. As a result, there was no shift of the TFCC to the ulnar side during ulnar flexion, and the measurable area was considerably smaller.

The PIV method was used to evaluate the velocity magnitude of the articular disc and ECU. PIV is used in fluid engineering to visualize the direction and velocity of a fluid. In recent years, it has been applied to medical technology and has been reported in various fields, such as analysis of blood flow inside aneurysms and research on the relationship between airflow velocity and development involving the vocal cords [21,22]. In the field of musculoskeletal research, dynamic analysis using the PIV method was reported by Kawanishi et al. [10]. They used the PIV method to record glide of the vastus lateralis muscle and subcutaneous tissue after proximal femur fracture surgery. They then evaluated the relationship between gliding and pain during weight-bearing and knee-joint movements. Their results showed that there was a significant correlation between gliding and pain during weight-bearing. Among their measurements, the intra-rater reliability was 0.92, and the inter-rater reliability was 0.83, which demonstrates high reproducibility. Dilley et al. also used this method to measure longitudinal median nerve movements [11]. They used the cross-correlation method on US images to measure the longitudinal movement of the median nerve during wrist extension and index finger extension. As a result, the authors reported that US imaging estimates were reliable in repeated studies and were consistent with those obtained from cadaveric studies.

In the present study, the PIV method was used based on these studies. Our intra-rater reliability was 0.97, and inter-rater reliability was 0.89, indicating high reproducibility. In our results, the velocity magnitude of the articular disc was larger, especially in the longitudinal direction in the injury group. The velocity magnitude of the ECU was also increased in the injury group. This suggests that, in Palmer type 1B injury, the function of the articular disc, which contributes to stability, is disrupted. The articular disc is sandwiched between the volar and dorsal sides of the radioulnar ligaments and acts as a static stabilizer for the DRUJ. The deep attachment of the articular disc to the ulnar fovea is called the ligamentum subcruentum, which is particularly important for DRUJ stability [23,24]. In a cadaveric study of TFCC, it was reported that the dorsal/palmer translation doubled when the articular disc was dissected from the fovea [8]. In this model, the instability was further aggravated when the subsheath of the ECU was removed, suggesting that the ECU compensates for the failure of the articular disc stability. In the present study, the velocity magnitude of the ECU was significantly increased in the Palmer type 1B TFCC injury group compared to that in the control group. This may be due to the increased load on the ECU as compensation for the disruption of the stability function of the articular disc. In the Palmer type 1B TFCC injury, it is considered that the instability of the DRUJ is caused by the disruption of the function of the articular disc and the ECU tendon as a stabilizer.

Object tracking analysis using US images presents the problem of loss of sight. However, by using the cross-correlation method, that problem is reduced when compared to its presence through the use of other methods, such as particle tracking velocimetry or optical flow [25]. At present, US is a useful method for non-invasive measurement of locomotor movements, and PIV analysis of US moving images has the potential to be developed in the future.

This study has some limitations. First, only Palmer type 1B injury was observed for the TFCC injury group, and evaluation was not performed based on the difference in injury type. Although Palmer type 1B injuries are often treated surgically, they are often undiagnosed, due to the difficulties associated with the current diagnostic options, leading to chronic pain. Thus, the accuracy of the diagnosis of Palmer type 1B injuries was evaluated in this study. Second, no comparative study using a cadaver model was conducted. In future studies, further evaluation of the other types of TFCC injury will make it possible to show the usefulness of US as an auxiliary diagnostic modality for TFCC injury. Third, this study is a preliminary study, and the sample size is not large. In this study, patients who underwent surgical treatment were included for grouping based on the diagnosis. In actual clinical practice, many patients suffer from chronic pain, due to a lack of appropriate diagnosis and treatment; thus, we hope that further studies will promote the use of US-assisted diagnosis for TFCC injuries. Finally, although dynamic evaluation by cross-correlation method can obtain relatively high reproducibility, anisotropy and technical reproducibility problems still remain in US images. We hope that this method will be widely used in clinical practice and that diagnosis using US imaging will be further developed.

## 5. Conclusions

The observation of TFCC from using US images has recently been reported. US is inexpensive, minimally invasive, and is expected to be useful as an auxiliary tool for diagnosis. In our study, a relatively high reproducibility could be obtained by using a long-axis image parallel to the TFCC as a quantitative diagnosis. By analyzing the dynamics of the articular disc and the ECU tendon as the stabilizer of the DRUJ, observed pathologic changes may lead to the diagnosis of TFCC injury. After future comparisons using other injury-type patient groups, we hope that this method will help diagnose TFCC injury.

## Figures and Tables

**Figure 1 sensors-22-00345-f001:**
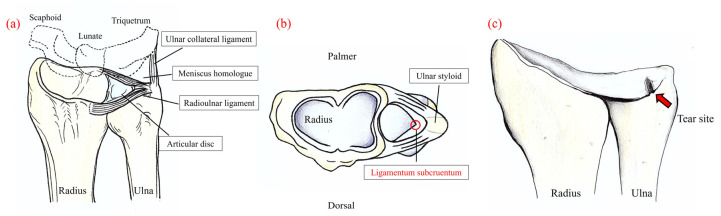
(**a**) Illustration of the main structures of the triangular fibrocartilage complex (TFCC). (**b**) Axial view. The ligamentum subcruentum is reported to be important for the stability of the wrist. (**c**) Tear site of the TFCC (Palmer type 1B).

**Figure 2 sensors-22-00345-f002:**
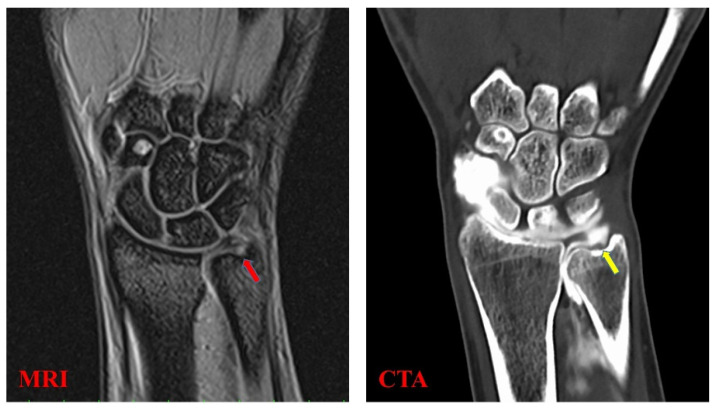
TFCC injury (Palmer type 1B) delineation via magnetic resonance imaging (MRI) and computed tomographic arthrography (CTA). MRI T2* imaging shows a high-intensity line on the disc of the TFCC (red arrow). Moreover, CTA pooling of the contrast medium at the TFCC attachment site of the ulnar head is visible (yellow arrow).

**Figure 3 sensors-22-00345-f003:**
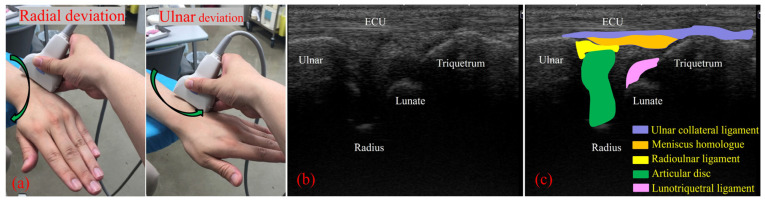
(**a**) Ultrasonography (US) transducer was positioned between the ulnar styloid process and the triquetrum, parallel to the extensor carpi ulnaris (ECU) tendon, with a 30 degree ulnar tilt from the vertical line. With this view, the wrist joint was moved from 15° of radial flexion to 25° of ulnar flexion. (**b**) US imaging of the TFCC by using the long-axis approach. (**c**) With this view, we can observe the ECU tendon, ulnar collateral ligament (light blue shade), meniscus homologue (orange shade), radioulnar ligament (yellow shade), articular disc (green shade), and lunotriquetral ligament (light purple shade).

**Figure 4 sensors-22-00345-f004:**
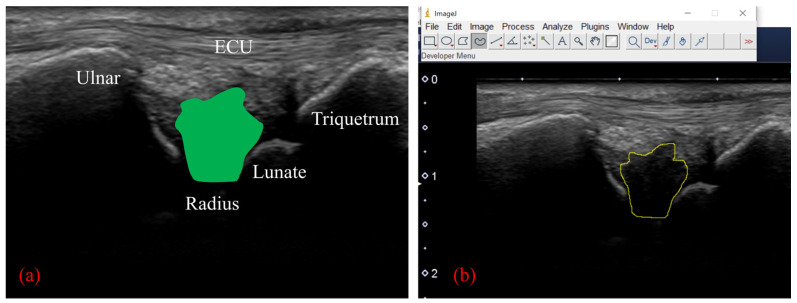
(**a**) We calculated the area of the articular disc up to the depth of the lunate cartilage (green area). (**b**) Calculation of the area using ImageJ software. The low-echo articular disc was manually plotted and measured with a measurement tool of Image J.

**Figure 5 sensors-22-00345-f005:**
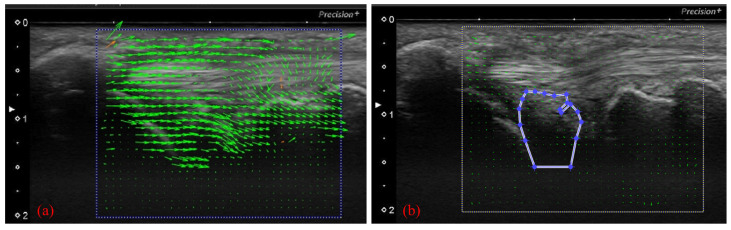
(**a**) Analyzing velocity magnitude image by using particle image velocimetry (PIV) fluid measurement software. With the US images, setting the articular disc and ECU tendon as regions of interest of the PIV. (**b**) Specification of the area to be measured (articular disc).

**Figure 6 sensors-22-00345-f006:**
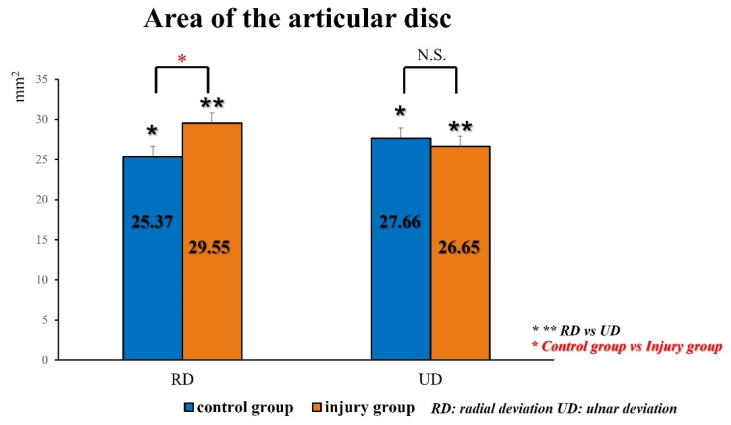
In contrast to the control group, the areas during radial deviation were statistically larger than those during ulnar deviation in the injury group.

**Figure 7 sensors-22-00345-f007:**
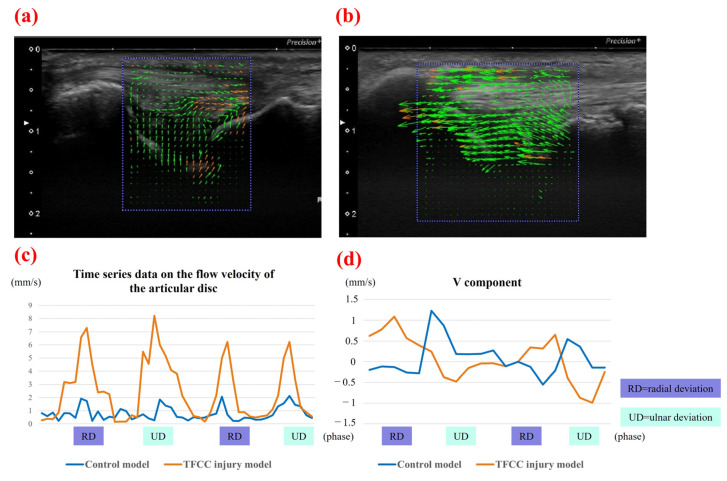
Flow PIV fluid measurement software of (**a**) the control model and (**b**) the TFCC injury model. (**c**) Representative data of the time series data on the velocity magnitude of the articular disc. Flow velocity was significantly higher in the injury group during both radial and ulnar deviation. (**d**) Representative data of the v-component. The v-component is moving in different vectors in the control group and the injury group.

**Figure 8 sensors-22-00345-f008:**
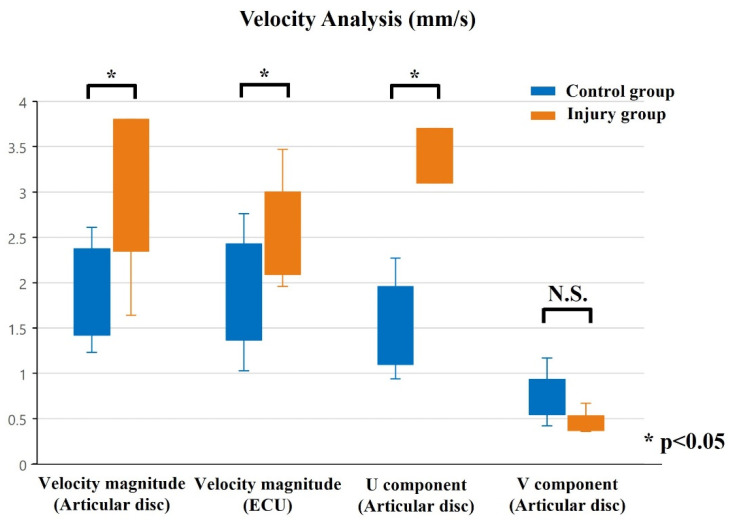
Results from PIV analysis. In the injury group, the velocity magnitude of the articular disc and the ECU tendon was significantly higher than that in the control group. The u-component of the articular disc was also significantly higher.

## Data Availability

The data presented in this study are available upon request from the corresponding author. The data are not publicly available, due to confidentiality concerns.

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
