# Peer review of "Motion Analysis of Triangular Fibrocartilage Complex by Using Ultrasonography Images: Preliminary Analysis"

_sensors, 2022, doi:10.3390/s22010345_

Round 1

Reviewer 1 Report

The authors present their findings on diagnosis of TFCC injury via US.  An experiment was designed to show the coexist of articular disc velocity and distal ratioulnar joint injury. The test volumn was small. The methodology is short of originality. This work is interesting and might serve as the preliminary result of a long pipeline for motion analysis, but not good enough for a journal paper.

Reviewer 2 Report

The manuscript submitted by Shinohara and co-authors proposes a tool to study the motion of the triangular fibrocartilage complex (TFCC) through ultrasound images. The  manuscript is interesting but written in a very technical way that expects a very specific expertise of readers (“… the ligamentum subcruentum, which is particularly important for DRUJ stability …”) and in my opinion would escape the interest of the average reader of Sensors unless it is adapted accordingly.

The introduction is just too short and assumes expertise in terms of anatomy of the TFCC, its complications, the imaging, Palmer type 1B even the 2019 protocol. Sensors has a wide readership that spans acquisition modalities and signal processing, but still it is important that the manuscript is clear and useful to a wide variety of readers. The manuscript seems written for experts in these particular type of injuries. Thus, I suggest to expand significantly the introduction and include:

An illustration of the TFCC and its anatomy articular disc, meniscus homologue, and juxta-articular ligaments.

An illustration or description of TFCC injuries and why the diagnosis of TFCC injury remains difficult

An illustration of the TFCC zone with several imaging modalities like xrays, CT and MRI. The justification to use US over any other modality is clear, but still it would be good to compare a few examples for better understanding of the problem.

In Materials and methods, please illustrate the data, and in particular what Palmer 1B and not palmer 1B.

“While under these views, the wrist was actively moved from 10 degrees of 90 radial deviation to 20 degrees of ulnar deviation at a 60 rounds per minute rhythm …” please explain why this was done and if possible illustrate..

The methodology seems reproducible and well described and the results are interesting showing a clear statistical difference between controls and injuries.

The discussion provides insights into the study including limitations, however, it has not been said why only Palmer 1B were selected.

Reviewer 3 Report

1. Musculoskeletal ultrasound allows physicians to see—in high resolution—a person’s muscles, tendons, ligaments, nerves and joints. It can focus on the exact location of the patient’s pain and use dynamic imaging to determine how motion affects the area of concern. The purpose of this study was to quantitatively analyze TFCC by performing motion analysis using Ultrasound. Please give the novelty of this study.

2. Only 27 cases were chosen for this study, please give the power analysis of this study.

3. The study is based on measurements, please give the Error analysis for the measurements.

4. How to solve the occlusion problem in echo data?

5. Lacks of comparison with other methods, such as MRI.

6. Please give the uncertainty of the parameters in this statistical model. 

Round 2

Reviewer 3 Report

accept

Author Response

This manuscript is a resubmission of an earlier submission. The following is a list of the peer review reports and author responses from that submission.